# Examining the Impact of Knowledge Mobilization Strategies to Inform Urban Stakeholders on Accessibility: A Mixed-Methods study

**DOI:** 10.3390/ijerph17051561

**Published:** 2020-02-28

**Authors:** Delphine Labbé, Atiya Mahmood, William C. Miller, W. Ben Mortenson

**Affiliations:** 1Department of Occupational Science and Occupational Therapy, University of British Columbia, Vancouver, BC V6T 1Z4, Canada; bill.miller@ubc.ca (W.C.M.); ben.mortenson@ubc.ca (W.B.M.); 2Rehabilitation Research Program, Vancouver, BC V5Z 2G9, Canada; 3Department of Gerontology, Simon Fraser University, Vancouver, BC V6B 5K3, Canada; amahmood@sfu.ca; 4ICORD Research Centre, Vancouver, BC V5Z 1M9, Canada

**Keywords:** disability, urban, accessibility, participatory research, knowledge translation, evaluation

## Abstract

Urban areas offer many opportunities for people with disabilities, but limited accessibility may prevent their full engagement in society. It has been recommended that the experience-based perspective of people with disabilities should be an integral part of the discussion on urban accessibility, complementing other stakeholder expertise to facilitate the design of more inclusive environments. The goals of this mixed-method study were to develop knowledge mobilization (KM) strategies to share experience-based findings on accessibility and evaluate their impact for various urban stakeholders. Using a participatory approach, various KM strategies were developed including videos, a photo exhibit and an interactive game. These strategies were evaluated based on various impact indicators such as reach, usefulness, partnerships and practice changes, using quantitative and qualitative methods. The findings suggested that the KM strategies were effective in raising the awareness of various urban stakeholders and providing information and guidance to urban planning practices related to accessibility.

## 1. Introduction

Over one billion people live with disability worldwide [1]—15% of whom live in urban areas [2]. In Canada, in 2017, 2.7 million people reported living with a mobility disability [3] and 35% lived in large urban municipalities [4]. Urban areas offer a variety of opportunities and services for people living with disabilities, but the lack of accessibility may prevent them from fully engaging in the society [5,6]. Accessibility, defined as “the equal opportunity to make use of goods and benefits and to participate in ordinary, common life according to one’s preferences” [2] (p. 3), directly influences how people with disabilities are perceived, treated and included in their communities [6,7]. The United Nations Urban Agenda made it a priority in 2017 to promote measures that facilitate equal access to public spaces, facilities, technology, systems, and services for persons with disabilities in urban settings [8]. 

Globally, policy makers and urban planners have made efforts to increase accessibility in recent decades. Countries such as the United States, United Kingdom, France and Australia have put in place accessibility related laws and policies to regulate buildings and the public realm. Despite the positive impacts of these regulations [6], concerns continue to be raised about how the design of public spaces fails to address the needs of people living with disabilities [9,10]. Several authors have suggested that decision makers and urban planners may lack necessary knowledge about accessibility and disability related issues to design public spaces that are fully accessible and inclusive [5,11,12]. Therefore, it is crucial to include the perspectives of people with disabilities in any discussion on urban accessibility [10,13,14].

Including people with disabilities in the planning process has several positive implications. People with disabilities bring experience-based knowledge [15,16] that complements the practical and political expertise of urban planners and decision makers [16,17]. People with disabilities could, for instance, explain how certain accessibility legislation requirements influence their daily experience or offer insights about the qualities of the built environment that goes beyond building standards [18]. This exchange of perspectives can strengthen policy planning and implementation [19,20]. Indeed, public participation that emphasizes empowerment, trust and learning has been associated with the design of urban spaces of higher quality [21]. Despite the potential benefits of public involvement, people with disabilities are still not regularly included in urban planning decision making [22] and the best ways to share and integrate their knowledge and experiences are not yet clear [13]. 

### 1.1. Knowledge Mobilization

Knowledge mobilization (KM) strategies could be used to inform urban planners and decision makers about the needs and embodied experiences of the people with disabilities [23]. KM is an umbrella term, primarily used by social science researchers, that describes a large category of strategies encompassing the full process of generating, accessing, sharing, and using information [20] (many other terms exist such as knowledge transfer or knowledge utilization; see [24] for a discussion of those terms). KM is intended to reduce the gaps between the knowledge users and relevant empirical knowledge, and to develop actions based on that knowledge [24,25]. Research has suggested that providing decision-making stakeholders with timely access to relevant evidence-based knowledge helps to target users’ needs more effectively [26,27,28]. KM has been used to inform practices of various stakeholders in studies focusing on public health initiatives for people with and without disabilities [19,29] and on environmental management [27,30]. However, very few studies have focused on KM for urban planners and decision makers regarding accessibility [12,17,31]. Interviews with urban planners suggested that through knowledge exchange and public consultations, they perceived they would get a better understanding of the accessibility terminology and the needs of people with disabilities [17]. 

KM is an iterative social process [24,32]. Different phases of KM include knowledge gathering, creation of KM strategies, and evaluation of the KM impacts [32,33,34]. Lavis and his colleagues [35] outlined five fundamental evidence-based principles of KM: (1) understanding the target audience (who?); (2) tailoring messages and practices to suit the audience (what should be transferred?); (3) using credible messengers (shared by whom?); (4) using effective methods for conveying messages (what is the process and supporting communication infrastructure?); (5) measuring the impact of KM (how is it evaluated?). Evaluation has been identified as a key component of KM, because it allows knowledge use to be monitored, and it can be used to demonstrate the impact of KM for the targeted users and settings; however, it has often been overlooked [26,33,35]. 

Based on a systematic review, Fazey and his colleagues [34] identified five principles for KM evaluation. First, they highlighted the importance of developing an evaluation that could apply to multiple end users. The second principle was to identify the KM goals and expected outcomes prior to the development of the KM strategies as it allows the creation of evaluation measures that are congruent with the goals [36]. The third principle emphasized the importance of evaluating both the KM process and outcomes. For instance, documenting the process could provide pertinent information to understand the KM outcomes [24,33]. Making the evaluation an integrated part of the KM process constituted the fourth principle. Finally, the authors [34] suggested using mixed-methods to conduct the evaluation. Qualitative methods could help capture social aspects of KM (e.g., practices or experiences) [34] and bring a nuanced understanding of the process and outcomes [11] whereas quantitative approaches could be used to measure knowledge uptake and its relationship to intended outcomes [37]. The application of these principles has been emphasized particularly for interdisciplinary KM and more complex issues [34], such as accessibility in the urban environment, which involves many components and multiple stakeholders. 

### 1.2. Aim

To date, no research has examined the impact of different methods for delivering evidence-based knowledge to improve accessibility and inclusion of people with disabilities. Therefore, this project aimed to describe the creation of different KM strategies to share the embodied experience and knowledge of people with disabilities on accessibility issues to different urban stakeholders and to evaluate the perceived impact of these strategies to inform urban planning practices in accessibility.

## 2. Methods

The development and evaluation of KM strategies represented the last phase of a large community-based participatory project (CBPR), entitled “*Enabling Mobility And Participation among those with Disabilities* (dEMAND)”. This multi-year mixed-method study [38] was conducted with 94 people using mobility devices (MD) (e.g., wheelchairs, scooters, walkers, canes and crutches) in two Canadian provinces: British Columbia (BC, *n* = 60) and Quebec (Qc, *n* = 34). Three cities in BC (due to local context) and one in Qc were involved in the project. The chosen cities shared similarities but were also complementary. They were all characterized by a mix of high-rise residential buildings and mixed-use development in their urban centres as well as by having suburban areas that included residential housing, light industry and commercial centre, with semi-rural areas at the edges. The two provinces differed in terms of climate (e.g., moderate vs. extreme) and weather conditions (e.g., mostly rain vs. heavy snow in the winter season). The study used various CBPR methods including GPS tracking and activity diary, photovoice and participant-led environmental audits to explore the experiences of MD users in the urban environment. A baseline survey was also used to collect demographic and other information (e.g., mental health, confidence with negotiating the environment). A community advisory committee composed of people with disability, city stakeholders and staff of community-based organizations informed and guided the project in each province. This project received approval from the local University research Ethics boards. The paper focuses on the development of the KM strategies and the evaluation component for the BC sites only. 

### 2.1. Data Collection

#### 2.1.1. Development of the KM Strategies

The integrated KM phase’s overall goal was to inform and increase awareness of decision makers, urban stakeholders and the public about accessibility issues for people with disabilities, especially MD users. Additionally, the team aimed to start a dialogue around these issues and co-identify potential solutions to foster the development of inclusive and accessible communities. The KM strategies were developed based on the five principles identified by Lavis and colleagues [35]. The strategies created focused on municipal staff and decision makers, as well as community-based organizations and the public (first principle: target audience). This decision emerged after a discussion with the community advisory committee members. The team then tailored the messages of the KM strategies (second principle) to suit that audience, especially by highlighting the embodied experience of MD users collected during the project. Based on the CBPR approach, the messengers (third principle) were both the MD users and the academic research team. Following the fourth principle, the team developed a variety of KM strategies using different but complementary media. Making use of a diversity of media has been found to increase the uptake of knowledge [34]. Thus, the KM strategies developed included (1) a video series, (2) a photo exhibit, and (3) an interactive game on accessibility. The details of these KM strategies are presented in the results section. Finally, the team developed and implemented an evaluation of the KM impact (fifth principle), presented below.

#### 2.1.2. Evaluation of the KM Strategies

We used a mixed-method approach to evaluate the KM process and outcomes, because it allowed for a better understanding of the KM evaluation results [11,34,36]. We developed the evaluation strategies based on The Knowledge Translation Planning Template (KTPT) [39] and the five principles of KM evaluation described above [34]. The KTPT was created to assist in the development of KM plans. The KTPT provided a list of questions to guide the evaluation, including “What internal/external factors do you need to consider?”; “Will methods be quantitative, qualitative or mixed?”; “Is the evaluation look at the process or the outcomes, or both?” The KTPT also provided a list of 10 KM impact indicators, among which, the team which the team selected the following based on the KM goals: *reach*, *usefulness*, *use*, *partnership/collaboration, and practice changes*. Table 1 presents the KM evaluation indicators and the methods used to evaluate them.

The methods used for the evaluation included (1) journaling, (2) observations and a (3) questionnaire with open and closed-ended questions. (1) Using a journal has been recommended by Barwick [39] to document more quantitative indicators such as the number of participants, the number of downloads or the number or products/services developed by the participants. The first author completed the journal. (2) Observations were used to document how stakeholders reacted to and understood the knowledge being shared, as suggested by previous KM evaluation studies using a CBPR approach [40,41]. The observations were recorded as field notes and included the first author’s observations and well as those shared by other research team members and student volunteers who help sharing the KM strategies. (3) Finally, a questionnaire was developed to evaluate the indicators of usefulness and use of the KM strategies. The questionnaire, available in paper and online format, was adapted from previous KM studies with similar populations [26,42]. It included seven multiple choice questions (e.g., The [name of the activity] enabled me to increase my knowledge about environmental barriers encountered by MD users), two open-ended questions (“What information/element[s] of the [name of the activity] was [are] the most useful?” and “What element[s] has [have] not been covered by these activities?”) and three demographic questions (gender, age and occupation). 

### 2.2. Data Analyses

Descriptive statistics were conducted on the closed questions of the questionnaire. One-way ANOVA was used to compare the questionnaire findings across the KM strategies and the KM participant. Following an inductive approach, the answers to the open-ended questionnaire items as well as the observations were analyzed using a conventional content analysis approach [43]. The first author created the initial codes through a close reading of the materials, and then grouped those codes into categories. She then discussed these categories with the second author to establish final labels and definition of the categories. Finally, the entire qualitative data set was reviewed again for consistency and completeness. Procedures were established to promote trustworthiness [44]. For instance, to enhance credibility, the analysis followed an iterative process involving a back-and-forth movement between the transcriptions and the codes, as well as a validation by more than one researcher. The triangulation of methods (e.g., journal, observations and surveys) and of investigators also support trustworthiness.

## 3. Findings

### 3.1. KM Strategies Development and Sharing

#### 3.1.1. A Video series

A series of three videos (see Figure 1a) that highlights key issues regarding accessibility in the three BC cities was created: Vancouver, North Vancouver and New Westminster (https://www.youtube.com/watch?v=Aq0gplsrxnY&list=PL5JuRZWyyYpIBKOPsEv-gUTggfGvUz4vJ). We used videos because they convey a general idea more easily and clearly than other traditional media and facilitate retention of intended knowledge [45]. Following a CBPR approach, the research team and six dEMAND participants featured in the video worked with a professional videographer to identify the key messages to be highlighted in the videos. Those key messages were based on the research data from the participant-led environmental audits [38,46]. The videos’ protagonists also provided input during filming and feedback on the final versions of the videos in which they participated. In each video, two participants (one man and one woman, of different ages and using different MD) demonstrated the different challenges they experienced in their daily lives, as well as improvements being made or desired to support their inclusion in their respective communities. The videos were filmed to be complementary to each other; each video presented issues specific to each city but was also made to be relevant to the other cities. Each video lasted, on average, 4 min.

The videos were made available on YouTube, as well as shared electronically via email and e-newsletters and different social media platforms (e.g., Facebook, Twitter). The videos were also presented at city and citizens advisory committee meetings (e.g., older adults or people living with disabilities).

#### 3.1.2. Photo Exhibit

A photo exhibit was developed based on the photovoice data, following the participatory process of this methodology [47]. The details of the development and analysis of the photovoice are presented in another paper [48]. Eight project participants, one member of the advisory committee living with disability and the academic research team chose the pictures for the exhibit. Twenty-seven pictures, grouped under five themes, that illustrated the mobility and accessibility issues experienced by the participants were printed on large and portable boards accompanied with relevant participants’ quotes. The first theme, “En Route,” concerned the usability and safety of the physical path to reach a destination. Thresholds referred to transition spaces being limited or having problematic access. The third theme, Temporal Rhythms, highlighted that accessibility fluctuates with daily and seasonal variations as well as people’s action while the fourth theme, Making Change Happen, referred to the agency of the MD users negotiating their environment and improving it. Finally, The Paradox of Accessibility, showcased how places and features could simultaneously be facilitators and barriers for people with disabilities. At least one picture from each of the three municipalities was presented for each theme. Figure 1b shows the photo exhibit display and one of the boards. The photo exhibit was presented as part of different community outreach events.

#### 3.1.3. Interactive Game

The final KM strategy was an interactive board game entitled, “On the Move Participation and Inclusion”, and was developed in collaboration with one advisory community committee member with lived experience with MD and expertise in accessibility and transportation issues. The game content was based on the photovoice and the participant-led environmental audit findings as well as a literature review on neighborhood factors influencing MD users [46]. The board game was designed to illustrate the daily experiences of MD users within an urban context (see Figure 1c). Each player assumes the identity of a MD user (e.g., a young woman, using a manual wheelchair, who studies at the university) and has to get to as many destinations as possible within a fictitious city while encountering and evaluating mobility barriers and facilitators within the game time frame. During and after the game, the players are encouraged to take part in a dialogue about the mobility related features of the environment from the perspectives of diverse characters of varying age and levels of mobility. The interactive game was played during the community outreach events. Members of the research team, including students and advisory board members with disabilities, facilitated the game. 

### 3.2. KM Evaluation

#### 3.2.1. Journal and Observations

Table 2 presents, for each KM strategy, the evaluation indicators of reach, usefulness, use, partnership and collaboration, as well as practice change, from the journal and observations. In terms of reach, the research team sent out direct invitations to a variety of stakeholders including the participants, city employees and decision makers, community-based organizations for people with disabilities and seniors, caregivers, students, health care professionals and the public. Indirect invitations were also sent through e-newsletters and posted on the websites of the project cities, community-based organizations, professional organizations of urban planners, and universities involved in the project. The KM strategies had coverage in traditional (e.g., local e-newspaper) and social media such as Twitter and Facebook. In terms of usefulness, the KM strategies were shared at different presentations and events. The KM strategies were shared as part of four community outreach events: two organized by the research team, one by a city partner and one by a professional organization of urban planners. A total of 236 participants took part in these four events including: 12 researchers, 23 students, 16 community-based organization staff, 112 municipal employees or decision makers, 28 citizens with disabilities (including eight dEMAND participants), 45 people i.e., six caregivers, seven health care workers and 32 others. Furthermore, following an invitation by participants at the outreach events, the research team made presentations to four city teams (49 city employees or officials), two local citizens advisory committees (seven older adults; 12 people with disabilities; one health professional) and one community-based organization (seven older adults). 

Valuable partnerships and collaborations emerged because of our KM (see Table 2). For instance, the researchers and MD users from the study were invited to be part of citizen committees for seniors and people with disabilities to share their experiences and knowledge on accessibility. An employee of a public transit organization who attended one of the community events also reached out to one of the video protagonists to consult with her about new public transit training for people with disabilities. In another instance, another city (not one of the original study sites) and a community-based organization reached out to one of the researchers to use the interactive game as part of an initiative to increase mobility and safety of older adults. Moreover, through the KM process, connections were created and trust was established between the research team, city staff and decision makers. These new collaborations led the city staff and decision makers to commit to change by participating in the development of interventions to increase accessibility. The KM process led to the creation of a large new partnership, which submitted a large multi-year research grant to pursue this collaboration. 

#### 3.2.2. Questionnaire

A total of 145 participants, 73 women and 55 men, completed the evaluation questionnaire for all the KM strategies. The questionnaire was completed by 34 city staff including engineers, urban and social planners, 17 people with disabilities, 16 students, 14 members of community organizations, seven rehabilitation professionals, seven family members and 17 others (e.g., public, members of advisory committees).

Table 3 presents the findings for the closed questions by each KM strategy. On average, each KM strategy received positive evaluations for both the usefulness and use indicators. No statistically significant difference was found in the evaluation between the strategies in terms of usefulness or use. On average, the participants agreed that the KM strategies were useful for all types of knowledge users’ groups (see Table 4). However, the KM strategies were perceived as more useful for urban planners (F (1,1433) = 6.569; *p* = 0.010) and social planners (F (1,1433) = 4.002; *p* = 0.046) than the other type of knowledge users, while they were judged as less useful for the people with disabilities (F (1,1433) = 30.273; *p* < 0.001) and the rehabilitation professionals (F (1,1433) = 15.299 *p* < 0.001). 

We identified different categories from the qualitative analysis of the open-ended questions and the field notes from the observations regarding the usefulness of each KM strategy (see Table 5). For the photo exhibit and the videos, participants noted that visuals were particularly useful. For instance, in describing the photo exhibit, a student said, “*The visual aid triggered an emotional response and made it very clear immediately what the barriers were and how easy it would be to fix [them]*.” (Female, 18–24 years old). Participants also considered that the videos and the photo exhibit were useful because the information was shared from the perspective of the MD users:
“*I think the most valuable aspect of this video is the various perspectives presented on the various access issues the two individuals confront, as well as their words. These help to point out the uniqueness of the individual needs of each user of a mobility device.*” (Female caregiver, 66 years of age)

The examples of tangible and real-life experiences depicted in the videos were perceived as particularly useful. According to participants, all three KM strategies improved their understanding of the daily mobility related challenges and barriers faced by MD users in the community. One urban planner noted, “[the game shows] *how some infrastructure can be both a facilitator and barrier depending on time*” (Female, 25–34 years old). Regarding the photo exhibit, a health care worker indicated, “*It illustrates barriers clearly that one may not think of or observe*” (Female, 45–54 years old). Fewer participants appreciated that the videos showed the positive aspects of the environment as well as potential solutions. The videos were also seen as raising awareness around accessibility and the needs of people with disability. As a city employee said, “*reframing of people’s experiences and range… Understand people with disability as active, engaged members of a community looking to get around and meet people*” (Male, 35–44 years old). Finally, the game was judged as useful because it was interactive and allowed the participants to put themselves in the place of someone using a MD. As a community-based organization employee emphasized, “*The interactivity of the game is the most engaging… To be able to actually digest how those issues may affect one’s life*.” (Male, 26–35 years old) The participants also described how the game allowed learning about accessibility issues in a playful manner.

Finally, the participants provided suggestions on how to improve the KM strategies. The main recommendations were: including a greater variety of disabilities such as people with hearing or visual impairment; providing more details about the challenges such as safety issues; placing more emphasis on changes that the cities had already implemented and the impacts of those changes. To a lesser extent, participants identified the importance of focusing on factors beyond the physical environment such as the stigma and attitude toward people with disabilities as well as the social policies. 

## 4. Discussion

This research was the first to focus on the development and evaluation of KM strategies to share the embodied experience and knowledge of MD users on accessibility issues to different urban stakeholders and to inform urban planning practices on accessibility to foster the development of inclusive communities.

Our findings suggest that using various interactive media in the KM strategies was effective to attain the study’s KM goals around urban accessibility and inclusion. KM is a social process [24] and these interactive KM strategies allowed face-to-face interactions, which have been shown to help persuade stakeholders of the importance of the knowledge shared [32]. Our findings exemplified this persuasion effect. For instance, following their participation in the KM strategies some of the municipal stakeholders wanted to change their practice as they started intervention projects with the research team following the KM. They also showed their intention to integrate more evidence-based knowledge in their policies by joining in a multi-year partnership grant application to continue to improve accessibility in urban environments. Moreover, the interactive KM strategies created a space for exchange that may be particularly important for people with disabilities as they are still often excluded from the decision-making process regarding issues that directly impact them, such as urban accessibility [9]. These interactive KM strategies also provided opportunities for municipal staff and decision makers to collaborate with new stakeholders including people with disabilities [31,49]. Promoting interaction among people with disabilities, the public and key stakeholders has been highlighted as one of the most effective ways to encourage positive attitudes toward people with disabilities and to foster the development of barrier-free environments [50]. Therefore, these exchanges should be encouraged and people with disabilities should be directly involved in the decision-making process of communities seeking to become fully inclusive [51]. 

Our findings revealed qualities that the various KM strategies held in common as well as unique impacts. On the one hand, participants evaluated all the KM strategies as effective. Moreover, the qualitative analysis of the open-ended questions and observations showed that all three KM strategies provided stakeholders with a better understanding of the environmental barriers faced by MD users. In addition, the videos and the photo exhibit were evaluated as giving the opportunity to see the accessibility issues from the eyes of people with disabilities. Understanding the perspective of people with disabilities allows for a better implementation of urban practices based on universal design principles [51] and deepens the understanding of the accessibility standards by urban stakeholders [18,49]. On the other hand, the videos were the only KM strategy evaluated as useful to raise awareness around mobility issues experienced by MD users. Raising awareness is crucial to strengthen the social participation of people with disabilities [12,31]. Moreover, Meagher and colleagues [11] argue that general awareness raising could lead to changes in practice and is a way that research can have a high impact. Our results seem to confirm this argument as stakeholders’ intention to change their practices were documented in the evaluation of the KM strategies. The game also had a unique impact. It was evaluated as allowing learning about accessibility issues in an entertaining way. This kind of playfulness is recommended as a means of contributing to learning and knowledge exchange [52]. Indeed, our findings showed that community-based organizations reached out to the researchers to use the game with other populations such as older adults to support open dialogues around inclusive urban environment. These findings suggest the importance of developing innovative KM strategies beyond traditional strategies, e.g., publications and conferences, to address social issues that involve many actors. These findings are in line with the literature advocating that the use of multiple methods to share knowledge contributes to different types of participation, knowledge uptake and changes in practices [19,34].

In understanding what types of knowledge are useful to whom, our study findings may inform future interventions and research projects on accessibility. These efforts may help respond to the needs of the people directly impacted by these issues, as well as those affected indirectly (e.g., caregivers). For instance, it may be helpful to understand what urban planners and other stakeholders need to know to create evidence-based strategies and action plans that respect and complement their own expertise [16,17,19]. In addition, using mixed-methods for the evaluation, as well as documenting the outcome and processes allowed us to capture both the social dimension of KM and to quantify some of the KM indicators [34,37]. Furthermore, the evaluation findings suggest that using a participatory approach allowed the creation of KM strategies that meaningfully conveyed MD users’ perspectives, while also informing the concrete practices of the urban planners and city officials. In addition, the participatory involvement of MD users in the design and implementation of the KM strategies appeared to provide users with a sense of ownership in the study findings [20]. For instance, the video protagonists were actively involved not only in the sharing that the KM strategy, but also actively shared the other KM strategies (and associated events) with their networks. Taking an integrated and participatory KM approach provided space for what Kothari [53] described as the rebalancing of what is considered, “expertise.” 

### Limitations and Future Research

The main limitation of this research was the cross-sectional nature of the evaluation of the KM strategies. The research team was able to highlight the immediate impact of the KM strategies; however, we do not know how these initiatives will impact policies and practices in the long term. Intervention studies and partnerships will continue these KM strategies, but longitudinal data collection would have allowed us to document the long-term outcomes of KM. Additionally, in-depth interviews with a few key participants from each type of knowledge user group may have provided a deeper understanding of the knowledge gained and of the expressed intentions to change. The type of KM strategies developed in this study were chosen by the research team, though the people with disabilities were involved throughout the development process. This could potentially be a limitation of this study, as they may have preferred to share their experiences in other ways, which they felt was more empowering [54]. However, our findings suggest that many groups of users, including people with disabilities, perceived the shared knowledge as useful. The participants in the KM evaluation were all English speaking and mostly from a Caucasian background. It would be interesting to conduct future studies on the usefulness of the KM strategies for populations from different cultural backgrounds as the intersections of ethnicity or gender have been identified as having an important influence on the experience of disability and mobility [55]. 

## 5. Conclusions

By highlighting key issues related to the daily mobility experience and social participation of people with disabilities, the KM strategies gave voice to a frequently overlooked group in urban planning. Making the experiences of people with disability explicit facilitated communication among different stakeholder groups around how to develop urban planning solutions that could address diversity and promote inclusive communities. The evaluation of the KM process helped to determine the effectiveness of various KM strategies to share valuable and relevant information to a variety of stakeholders. The development and implementation of the KM strategies started a dialogue between relevant stakeholders and initiated a reflective process that may lead to recognition of the needs of people with disabilities including MD users in the urban planning and design of our communities. Developing accessibility policies to create inclusive communities is a complex process that could benefit from thoughtful, varied and well-designed evidence-based KM strategies.

## Figures and Tables

**Figure 1 ijerph-17-01561-f001:**
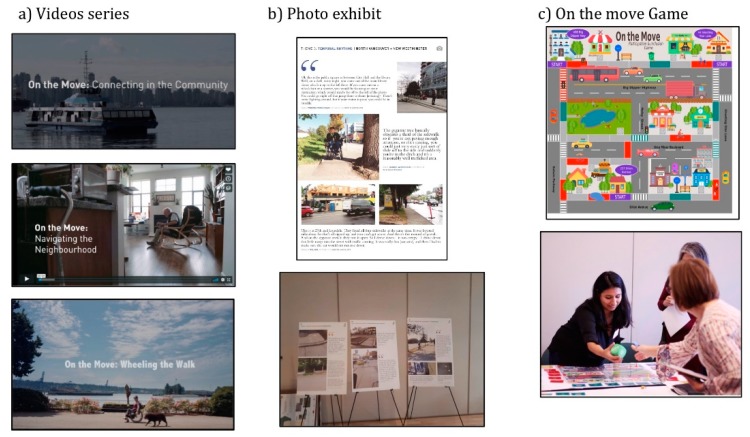
Knowledge Mobilization Strategies.

**Table 1 ijerph-17-01561-t001:** Mobilization Indicators and Methods Used to Assess Them.

Type of Indicators	Subtype of Indicators	Method
Reach	Number of invitations distributed	Journal
	Number of requests for the knowledge mobilization (KM) strategies	
	Number of downloads/hits	
	Media exposure (including social media)	
	Number of questionnaires completed	
Usefulness	Number of participants and duration/type of participation	Journal
	Number of participants who participated in development	
	Satisfaction with KM strategies	Questionnaire
	Usefulness of gained knowledgev	
	Changed views	
Use	Number of users adapting the information	Observations
	Number of people using the KM strategies to inform policy/advocacy/enhance programs, training, education, or research	
	Number of persons using the KM strategies to improve their practice	Questionnaire
	Intend to use	
Partnership Collaboration	Number of products/services developed, or disseminated by the participants	Journal
	Social network growth, influences, collaboration	Observations
Practice change	Intention or commitment to change	Observations

**Table 2 ijerph-17-01561-t002:** Impact Indicators from the Journal and Observations by KM Strategies.

Type of indicators	Subtype	Videos	Photo Exhibit	Interactive Game
Reach	Number distributed or invitations		In total, 1500 direct invitations via email In total, 11 email blast invitations sent through newsletters or online event-planning site	In total, 1500 direct invitations via email In total, 11 email blast sent through newsletters or online event-planning site
	Number of views	In total, 1411 views		
	Media exposure (include social media such as Facebook, Twitter)	Four Facebook posts In total, 11 Twitter posts (25 retweets)	One online newspaper article Two Facebook posts In total, 11 Twitter posts (41 retweets)	One online newspaper article Two Facebook posts In total, 11 Twitter posts (41 retweets)
Usefulness	Number of participants	Eight presentations with 76 participants Three community outreach events with 206 participants	Four community outreach events with 236 participants	Four community outreach events with 236 participants
	Duration of participation	Average view of 2 min 20s	Average of 25 min	Average of 20 min
	Type of participation	Passive	Interactive	Interactive
	Number of people who participated in the development of the KM Strategies	Six dEMAND participants	One member of the advisory committee Eight dEMAND participants	One member of the advisory committee
Partnership and Collaboration	Number of products/services developed, or disseminated with partners	One development of research project	Two invitations to present the KM strategies to other city events	Three requests to use the game in other contexts
	Social network growth, influences, new collaboration	Four new collaborations: two between MD users and city decision maker and staff; two between researchers and city staff	Two new collaborations between researchers and city staff	Two new collaborations between researchers and community-based organizations
Practice Change	Intent or commitment to change	Development of guidelines to change workers practice Intention to include the knowledge in accessibility strategy	Development of guideline to change workers practice Intention to include the knowledge in accessibility strategy	

**Table 3 ijerph-17-01561-t003:** Questionnaire Evaluation of the Usefulness and Use of KM Strategies.

Type of Indicators	Questions	Video (*n* = 134) Mean (SD)	Photo Exhibit (*n* = 75) Mean (SD)	Interactive Game (*n* = 70) Mean (SD)
Usefulness	The length of the [name of activity] is appropriate.	4.36 (0.77)	4.39 (0.77)	4.27 (0.83)
Usefulness	The content corresponds to my needs and interest.	4.30 (0.79)	4.42 (0.75)	4.37 (0.89)
Usefulness	The [name of the activity] enabled me to increase my knowledge about environmental barriers encountered by mobility device users.	4.30 (0.86)	4.39 (0.88)	4.43 (0.84)
Usefulness	This [name of the activity] changed my perspective on the experiences of mobility device users in their environment.	3.96 (0.99)	4.23 (0.89)	4.17 (0.96)
Use	I intend to apply the knowledge I got from [name of activity].	4.17 (0.83)	4.25 (0.82)	4.26 (0.87)
Use	I would share this [name of the activity] with my network.	4.30 (0.85)	4.47 (0.78)	4.45 (0.84)
Use	This kind of [name of the activity] is very useful for me and/or my work.	4.20 (0.84)	4.31 (0.85)	4.24 (1.00)

Note. Means and standard deviations are measured on a 5-point scale from 1 “strongly disagree” to 5 “strongly agree”.

**Table 4 ijerph-17-01561-t004:** Usefulness Evaluation of the KM Strategies by Audiences (*n* = 145).

Question	Urban Planners	Advisory Board Committee	Social Planners	Community Organization	Engineers	Rehabilitation Professionals	Students	People with a Disability	Researchers	General Public
	Mean (S.D)									
In my opinion, these activities would be useful for the following target audiences	4.55 * (0.85)	4.50 (0.84)	4.51 * (0.79)	4.40 (0.86)	4.50 (0.89)	4.07 ** (0.96)	4.31 (0.85)	3.99 * (0.99)	4.36 (0.88)	4.49 (0.78)

Note. Means and standard deviations are measured on a 5-point scale from 1 “strongly disagree” to 5 “strongly agree.” * *p* < 0.05; ** *p* < 0.01.

**Table 5 ijerph-17-01561-t005:** Qualitative Descriptive Categories by KM Strategies.

Categories	Videos	Photo Exhibit	Interactive Game
Using Visuals	x	x	
Seeing it from the MD users’ perspective	x	x	
Offering tangible, real-life examples	x		
Understanding better the barriers and challenges	x	x	x
Informing about the facilitators and possible solutions	x		
Raising awareness	x		
Interactivity and role playing			x
Entertaining			x

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
