# Peer review of "Examining the Impact of Knowledge Mobilization Strategies to Inform Urban Stakeholders on Accessibility: A Mixed-Methods study"

_ijerph, 2020, doi:10.3390/ijerph17051561_

Round 1
Reviewer 1 Report
This is an interesting use of mixed methods to share information about accessibility issues for mobility device users with various urban planning stakeholders. In general, I found the knowledge mobilization methods used of interest and engagement of people with disabilities in the process especially noteworthy. While the results were a bit ambiguous in terms of differential effectiveness with different stakeholder groups, the work is nonetheless worthy of consideration.
There are, however, several problems with the presentation that must be addressed before further consideration. First, there are multiple typographical errors and incorrect use of tense, etc. suggesting that English is not the primary language of the authors. The manuscript requires a good copy edit.
Second, the tables are very difficult to follow in large part due to poor formatting that results in many omissions of text. These need to be corrected and reformatted.
Third, the discussion section is far too long with much redundant information (particularly the 3rd and 4th paragraphs on p. 12. I suggest this section be tightened up.
Author Response
Thank you to Reviewer 1 for the comments. Please see below our response to these comments
First, there are multiple typographical errors and incorrect use of tense, etc. suggesting that English is not the primary language of the authors. The manuscript requires a good copy edit.
- A copy editing of the manuscript have been completed.
Second, the tables are very difficult to follow in large part due to poor formatting that results in many omissions of text. These need to be corrected and reformatted.
- The format of the tables changed widely with the formatting by the journal. Using the revised version provided, we reviewed all the tables to clarify them.
Third, the discussion section is far too long with much redundant information (particularly the 3rd and 4th paragraphs on p. 12. I suggest this section be tightened up.
- The 3rd and 4th paragraphs have been merged and tighten (see page 12). The overall discussion has been reviewed as well to improve the flow.
Reviewer 2 Report
1.The topic is fine, the structure is good, and the Mixed-Methods study is charming.
2.Some fresh papers should be cited, like the following:
Sun, H., Geng, Y., Hu, L., Shi, L., Xu, T., 2018. Measuring China’s new energy vehicle patents: a social network analysis approach. Energy. 153: 685-693. https://doi.org/10.1016/j.energy.2018.04.077.
Sun H., Bless Kofi E., Sun C., Kporsu A K. Institutional quality, green innovation and energy efficiency, Energy policy, (2019),135,111002. https://doi.org/10.1016/j.enpol.2019.111002.
3. For the conclusion part, some policy implications can be discussed.
Author Response
2.Some fresh papers should be cited.
- Thank you for the suggested papers.
For the conclusion part, some policy implications can be discussed.
- A sentence about policy implications have been added to the conclusion. See page 13
Reviewer 3 Report
The reviewed article is a valuable submission. It has significant practical value, but also considerable cognitive value, since it addresses an issue that is not well recognized in the literature. The article meets all the criteria of a valuable scientific work such as, an accurate introduction justifying the need to undertake analyzes based on a comprehensive understanding of the subject matter. In addition, the authors carefully and correctly outlined the assumptions of their own research and exhaustively described their procedure. The analysis of the authors' research results raises no objections, and the conclusions are accurate. The issues raised in the article are of great importance for the thematic profile of the journal and its international reach, since the described procedure is applicable in various socio-cultural conditions.
Expand report:
The reviewed article is a valuable submission. In their analyses, the authors focus on knowledge mobilization strategies that increase stakeholders’ knowledge about the availability of places and public space in urban environments. The stakeholders include both decision-makers responsible for shaping the accessibility of these environments, but also people directly interested in these places: people with physical disabilities. The use of the participatory method enabled researchers to explore the issue from the most optimal perspective of people with disabilities, which increases the accuracy of the findings and, as a consequence, the adequacy of the solutions designed on their basis. The article has significant practical value, but also considerable cognitive value, since it addresses an issue that is not well recognized in the literature. The issues raised in the article are of great importance for the thematic profile of the journal and its international reach since the described procedure is applicable in various socio-cultural conditions.
Detailed comments on the individual parts of the article:
Accurate introduction justifying the need to undertake analyzes based on a comprehensive understanding of the subject matter. Spelling mistakes in footnote 2 (p.2). The authors carefully and correctly outlined the assumptions of their own research and exhaustively described their procedure. The procedure is quite complex, which may make it difficult for the reader to isolate individual stages of the research process (problem design, data collection, and description of the results). I recommend the authors to review this matter. It is also worth explaining the reasons for choosing specific provinces and cities for the study. Additional information on the procedure and criteria for selecting the study group, which was very diverse would be helpful. This is important because the presented quantitative and qualitative analyses, aimed at evaluating the strategy, are key to the research goals. The analyzed differences refer to very unequal sample sizes. Why did the authors not ensure greater consistency in this respect?The strongest point of the article is its subject matter and the interesting strategies with a universal dimension. The weaker side is the selection of the examined group and the small sample sizes within subgroups. The stakeholder group should be diverse, but it would have been better to find more representatives of the subgroups.
Author Response
Thank you to reviewer 3 for their comments.
Accurate introduction justifying the need to undertake analyzes based on a comprehensive understanding of the subject matter. Spelling mistakes in footnote 2 (p.2).
- The spelling mistake has been corrected.
The authors carefully and correctly outlined the assumptions of their own research and exhaustively described their procedure. The procedure is quite complex, which may make it difficult for the reader to isolate individual stages of the research process (problem design, data collection, and description of the results). I recommend the authors to review this matter.
- The method and results have been reviewed to clarify the complex process of the data collection. We numbered the sections to highlight the parallel between the sections in method and the corresponding results. See page 3 to 11.
It is also worth explaining the reasons for choosing specific provinces and cities for the study.
- Explanation of the choice of the cities was added. See page 3
Additional information on the procedure and criteria for selecting the study group, which was very diverse would be helpful. This is important because the presented quantitative and qualitative analyses, aimed at evaluating the strategy, are key to the research goals. The analyzed differences refer to very unequal sample sizes. Why did the authors not ensure greater consistency in this respect?
- These knowledge mobilization (KM) strategies were shared as part of public invitations to a variety of stakeholders and despite the fact that we tried to have a good representation of each groups, we had no control of who did actually attend the public outreach events. Also, during those events, not all the participants saw all the KM strategies and we did not asked them to evaluate a strategies they did not interact with, which explain the inequality of the sample sizes. However, the homogeneity of variance and other assumptions for the statistical tests were all respected.
Round 2
Reviewer 1 Report
The authors have addressed my concerns. My only suggestion is further condensing of the discussion session for readability.